# Deconvolutional Networks on Graph Data

**Jia Li[1], Jiajin Li[2], Yang Liu[1], Jianwei Yu[2], Yueting Li[2], Hong Cheng[2]**
[1] Hong Kong University of Science and Technology
[2] The Chinese University of Hong Kong
`jialee@ust.hk`

## Abstract

In this paper, we consider an inverse problem in graph learning domain – "given the graph representations smoothed by Graph Convolutional Network (GCN), how can we reconstruct the input graph signal?" We propose Graph Deconvolutional Network (GDN) and motivate the design of GDN via a combination of inverse filters in spectral domain and de-noising layers in wavelet domain, as the inverse operation results in a *high frequency amplifier* and may amplify the noise. We demonstrate the effectiveness of the proposed method on several tasks including graph feature imputation and graph structure generation.

## 1 Introduction

Graph Convolutional Networks (GCNs) [6, 20] have been widely used to transform an input graph structure $A$ and feature matrix $X$ into graph representations $H$, which can be taken as a mapping function $f : (A, X) \rightarrow H$. In this paper, we consider its inverse problem $g : (A, H) \rightarrow X$, i.e., *given the graph representations smoothed by GCNs, how can we reconstruct the input graph signal?* More specifically, we initiate a systematic study of Graph Deconvolutional Networks (GDNs), to reconstruct the signals smoothed by GCNs. A good GDN component could benefit many applications such as reconstruction [44] and generation [19]. For examples in computer vision, several studies [45, 29] have demonstrated that the performance of autoencoders on reconstruction and generation can be further improved by encoding with Convolutional Networks and decoding with Deconvolutional Networks [46]. However, extending this symmetric autoencoder framework to graph-structured data requires graph deconvolutional operations, which remains open-ended and hasn't been well studied as opposed to the large body of solutions that have already been proposed for GCNs.

Most GCNs proposed by prior works, e.g., Cheby-GCN [6] and GCN [20], exploit spectral graph convolutions [33] and Chebyshev polynomials [15] to retain coarse-grained information and avoid explicit eigen-decomposition of the graph Laplacian. Until recently, [39] and [8] have noticed that GCN [20] acts as a *low pass* filter in spectral domain and retains smoothed representations. Inspired by prior works in the domain of signal deconvolution [22, 9, 13], we are motivated to design a GDN by an inverse operation of the *low pass* filters embodied in GCNs. Furthermore, we observe directly reversing a *low pass* filter results in a *high frequency amplifier* in spectral domain and may amplify the noise. In this regard, we introduce a spectral-wavelet GDN to decode the smoothed representations into the input graph signals. The proposed spectral-wavelet GDN employs spectral graph convolutions with an inverse filter to obtain inversed signals and then de-noises the inversed signals in wavelet domain. In addition, we apply Maclaurin series as a fast approximation technique to compute both inverse filters and wavelet kernels [8].

We evaluate the effectiveness of the proposed GDN with two tasks: graph feature imputation [35, 44] and graph structure generation [19, 14]. For the former task, we further propose a graph autoencoder (GAE) framework that resembles the symmetric fashion of architectures [29]. The proposed GAE outperforms the state-of-the-arts on six benchmarks. For the latter task, our proposed GDN can

35th Conference on Neural Information Processing Systems (NeurIPS 2021).

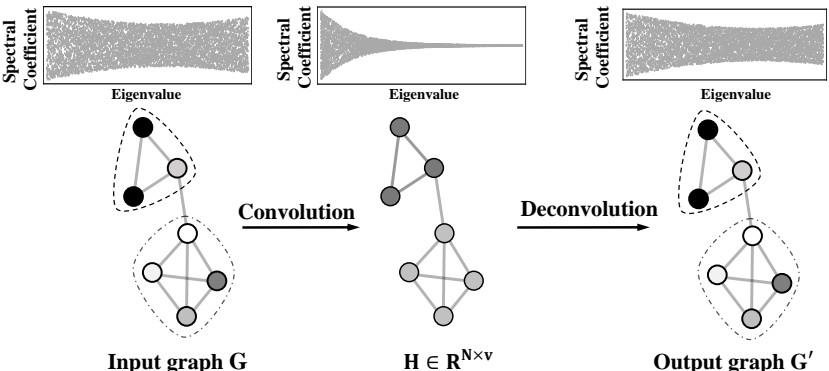

Figure 1: Schematic diagram of the graph autoencoder perspective. We plot graphs in spatial domain together with their signal coefficients in spectral domain. In the encoding stage, we run a GCN model to derive smoothed node representations. In the decoding stage, we reconstruct the graph signal via a GDN model.

enhance the generation performance of popular variational autoencoder frameworks including VGAE [19] and Graphite [14].

The contributions of this work are summarized as follows:

- We propose Graph Deconvolutional Network (GDN), the opposite operation of Graph Convolutional Network (GCN) that reconstructs graph signals from smoothed node representations.

- We study GDN from the perspective of inverse operations in spectral domain and identity the inverse operation results in a *high frequency amplifier* and may amplify the noise. To solve the noise issue, a de-noising process based on graph wavelet is further proposed.

## 2 Preliminaries

In this work, we are given an undirected, unweighted graph $G = (V, A, X)$. $V$ is the node set and $N = |V|$ denotes the number of nodes. The adjacency matrix $A \in \mathbb{R}^{N \times N}$ represents the graph structure. The feature matrix $X \in \mathbb{R}^{N \times d}$ represents the node attributes.

### 2.1 Convolution on graphs

The convolutional layers are used to derive smoothed node representations, such that nodes that are similar in topological space should be close enough in Euclidean space. Formally, the spectral graph convolution on a signal $x \in \mathbb{R}^N$ is defined as:

$$h = g_c * x = U \text{diag}(g_c(\lambda_1), \ldots, g_c(\lambda_N)) U^\top x, \tag{1}$$

where $\{\lambda_i\}_{i=1}^N$ are the eigenvalues of the normalized Laplacian matrix $L_{sym} = D^{-\frac{1}{2}} L D^{-\frac{1}{2}}$, $L$ and $D$ are the Laplacian and degree matrices of the input graph $A$ respectively, and $U$ is the eigenvector matrix of $L_{sym} = U \Lambda U^\top$. In this work, we focus on GCN [20], one of the popular convolution operations on graphs:

$$H = \text{GCN}(A, X), \tag{2}$$

where $H \in \mathbb{R}^{N \times v}$ denotes smoothed node representations. Specifically, [39] show that GCN is a *low pass* filter in spectral domain with $g_c(\lambda_i) = 1 - \lambda_i$, i.e.,

$$h_c = \sigma(U \text{diag}(g_c(\lambda)) U^\top x), \tag{3}$$

where $\sigma(\cdot)$ is the activation function.

## 2.2 Deconvolution on graphs

The deconvolutional layers are used to recover the original graph features given smoothed node representations. Formally, the spectral graph deconvolution on the smoothed representation $h \in \mathbb{R}^N$ is defined as:

$$x' = g_d * h = U\text{diag}(g_d(\lambda_1), \ldots, g_d(\lambda_N))U^\top h, \tag{4}$$

With this deconvolution operation, we can use GDNs to produce fine-grained graph signals from the encoded $H$.

$$X' = \text{GDN}(A, H), \tag{5}$$

where $X' \in \mathbb{R}^{N \times d}$ denotes the recovered graph features. We shall further discuss our design of GDN in Section 3.

From the perspective of encoder-decoder, the convolution operation could be used as an encoder and the deconvolution operation as a decoder. We show a schematic diagram of this encoder-decoder perspective in Figure 1.

## 3 Graph Deconvolutional Network

In this section, we present our design of GDN. Motivated by prior works in signal deconvolution [16], a naive deconvolutional net can be obtained using the inverse filter $g_c^{-1}(\lambda_i) = \frac{1}{1-\lambda_i}$ in spectral domain. Unfortunately, the naive deconvolutional net results in a *high frequency amplifier* and may amplify the noise [9, 13]. This point is obvious for $g_c^{-1}(\lambda_i)$ as the signal fluctuates near $\lambda_i = 1$ if noise exists.

### 3.1 Noise analysis

Our target is to recover the original graph features given smoothed node representations $h_c$ encoded by GCN [20]. Without the noise, it is naive to obtain the closed-form recovery relationship by the inverse operation $g_c^{-1}(\lambda) = \frac{1}{1-\lambda}$ directly, i.e.,

$$x = U\text{diag}(g_c^{-1}(\lambda))U^\top \sigma^{-1}(h_c), \tag{6}$$

Here, we assume the activation function $\sigma(\cdot)$ is an inverse operator, e.g., Leaky ReLU, sigmoid, tanh. Actually, it is quite reasonable to consider such a prototypical model of invertible networks in the theoretical analysis. Otherwise, we need to address an ill-posed inverse problem instead as we automatically lose some information (e.g., null space). Hence, it is impossible to recover the ground-truth even under the noiseless model, see [1] for details.

Graph noise is ubiquitous in real-world applications. Following the convention in signal deconvolution [9] and image restoration [13], let's consider the white noise model, i.e., $\hat{h}$ is the smoothed node representations mixed with graph convolution and white noise,

$$\hat{h} = \sigma(U\text{diag}(g_c(\lambda))U^\top x) + \epsilon, \tag{7}$$

where $\epsilon$ is the white Gaussian noise (i.e., $\epsilon \sim \mathcal{N}(0, \sigma^2\mathbb{I}_N)$).

Next, we show that it will amplify the noise if we apply the inverse operation to the white noise model directly. That is,

$$x' = U\text{diag}(g_c^{-1}(\lambda))U^\top \sigma^{-1}(\hat{h}), \tag{8}$$

**Proposition 1.** *Considering the independent white noise model (7) and (8), if the inversed activation function is deferentiable which satisfies $\partial\sigma^{-1}(0) \geq 1$, the variance of $x'$ is larger than the white noised $\epsilon$ added on the generative model.*

It was worthwhile mentioning that the condition regarding the inverse operator $\partial\sigma^{-1}(0) \geq 1$ is quite reasonable. Actually, it holds for many representative activation functions, including sigmoid, tanh and Leaky ReLU. Please refer to Appendix A for details. Proposition 1 shows the proposed inverse operation may introduce undesirable noise into the output graph. Thus, a de-noising process is necessary thereafter.

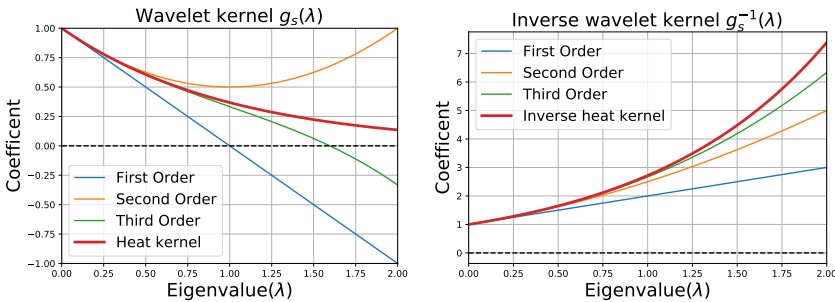

Figure 2: Low-order Maclaurin Series well approximate wavelet kernel and inverse wavelet kernel with $s = 1$.

## 3.2 The proposed GDN

To ameliorate the noise issue, we propose an efficient, hybrid spectral-wavelet deconvolutional network that performs inverse signal recovery in spectral domain first, and then conducts a de-noising step in wavelet domain to remove the amplified noise [28].

### 3.2.1 Inverse operator

Regarding the inverse operation in (6), the core is to explicitly compute eigen-decomposition. We propose to apply Maclaurin series approximation on $g_c^{-1}(\Lambda) = \sum_{n=0}^{\infty} \Lambda^n$ and get a fast algorithm as below:

$$U\text{diag}(g_c^{-1}(\lambda))U = \sum_{n=0}^{\infty} L_{sym}^n. \tag{9}$$

A detailed derivative of (9) can be found in Appendix E. When truncated low order approximation is used in practice with $n \leq 3$ [6], we derive a spectral filter with $g_c^{-1}(\lambda_i) = 1 + \lambda_i + \lambda_i^2 + \lambda_i^3$.

Following [29], we can use another activation function and trainable parameter matrix to estimate $\sigma^{-1}$, to avoid the potential non-invertible problem. Thus, the inverse operator becomes:

$$M = \sigma((I_N + \sum_{n=1}^{3} L_{sym}^n)HW_3), \tag{10}$$

where $W_3$ is the parameter set to be learned.

We contrast the difference between inverse operator in this work and other reconstruction methods including [47] and GALA [30].

**Inverse operator vs. GCN decoder [47]**  Compared with GCN decoder [47] which directly uses GCN for signal reconstruction, the proposed inverse operator demonstrates its efficacy in recovering the high frequency signals of the graph, as shown in Figure 3 (b). We shall further discuss this point in Section 3.3.

**Inverse operator vs. GALA**  GALA [30] can be viewed as the first-order approximation towards inverse operator in this work, with a spectral kernel of $1 + \lambda_i$. The difference is that our inverse operator can well approximate the inverse filter $g_c^{-1}(\lambda_i) = \frac{1}{1-\lambda_i}$ with third-order polynomials.

### 3.2.2 Wavelet de-noising

Wavelet-based methods have a strong impact on the field of de-noising, especially in image restoration when noise is amplified by inverse filters [28]. In the literature, many wavelet de-noising methods have been proposed, e.g., SureShrink [9], BayesShrink [5], and they differ in how they estimate the separating threshold. Our method generalizes these threshold ideas and automatically separates the noise from the useful information with a learning-based approach. Consider the input signal of wavelet system is mixed with Gaussian noise, the reasons why wavelet denoising works are: (1) a graph wavelet basis holds the majority of signal energy into a small number of large coefficients,

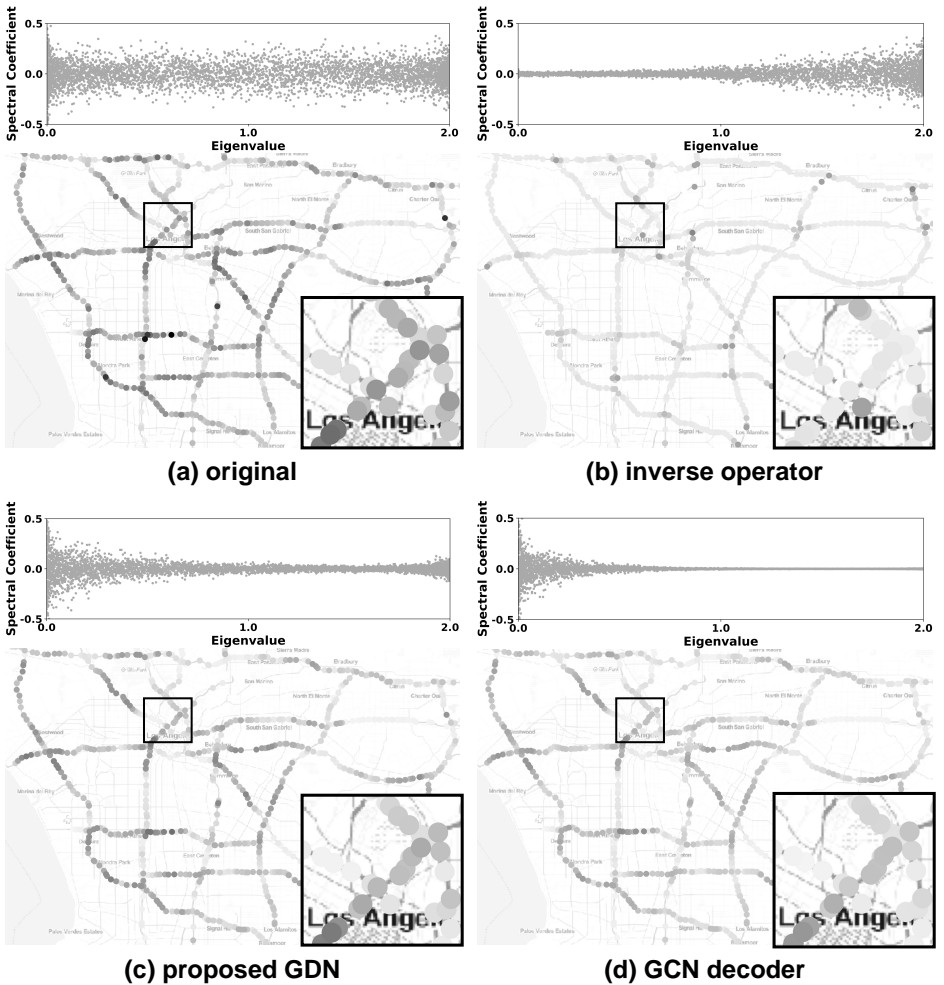

Figure 3: Illustration of road occupancy rates decoded by different approaches (zoom in to see details). A darker color implies a higher road occupancy rate. In each figure above, we plot the reconstructed signal in spatial domain together with coefficients in spectral domain. (a) the original, (b) inverse operator (RMSE = 0.11), (c) GDN/our model (RMSE = 0.07), (d) GCN decoder [47] (RMSE = 0.09).

and (2) Gaussian noise in the original graph domain is again a Gaussian noise in wavelet domain (with the same amplitude), see [16] for details. The core thus becomes how we can separate a small number of large coefficients and the noise in wavelet domain. Given the latter is unknown, we adopt a trainable parameter matrix to linearly transform the noise under zero and useful coefficients bigger than zero, where ReLU is used to eliminate the noise.

Consider a set of wavelet bases $\Psi_s = (\Psi_{s1}, \ldots, \Psi_{sN})$, where each one $\Psi_{si}$ denotes a signal on graph diffused away from node $i$ and $s$ is a scaling parameter [40], the wavelet bases can be written as $\Psi_s = U g_s(\Lambda) U^\top$ and $g_s(\cdot)$ is a filter kernel. Following previous works GWNN [40] and GRAPHWAVE [8], we use the heat kernel $g_s(\lambda_i) = e^{-s\lambda_i}$, as heat kernel admits easy calculation of inverse wavelet transform with $g_s^{-1}(\lambda_i) = e^{s\lambda_i}$. In addition, we can apply Maclaurin series approximation on heat kernel and neglect explicit eigen-decomposition by:

$$\Psi_s = \sum_{n=0}^{\infty} \frac{(-1)^n}{n!} s^n L_{sym}^n, \tag{11}$$

$$\Psi_s^{-1} = \sum_{n=0}^{\infty} \frac{s^n}{n!} L_{sym}^n, \tag{12}$$

To cope with the noise threshold estimation problem, we introduce a parameterized matrix in wavelet domain and eliminate these noise coefficients via a ReLU function, in the hope of learning a separating threshold from the data itself. Together with the representation $M$ raised in Section 3.2.1, we derive the reconstructed signal as:

$$X' = \Psi_s \text{ReLU}(\Psi_s^{-1} M W_4) W_5, \tag{13}$$

where $W_4$ and $W_5$ are trainable parameters.

We then contrast the difference between Wavelet Neural Network (WNN) in this work and other related WNNs including GWNN [40] and GRAPHWAVE [8].

**Scalability**  An important issue in WNN is that one needs to avoid explicit eigen-decomposition to compute wavelet bases for large graphs. Both GRAPHWAVE and GWNN, though trying to avoid eigen-decomposition by exploiting Chebyshev polynomials, still rely integral operations (see Appendix D in [40]) and there is no clear way to scale up. Differently, we use Maclaurin series to approximate wavelet bases, which has explicit polynomial coefficients and can resemble the heat kernel well when $n = 3$. Please refer to Figure 2 for more details.

**De-noising**  The purpose of both GWNN and GRAPHWAVE is to derive node presentations with localized graph convolution and flexible neighborhood, such that downstream tasks like classification can be simplified. On the contrary, our work implements wavelet neural network in the purpose of detaching the useful information and the noise amplified by inverse operator. Due to the different purpose, our work applies the activation function in wavelet domain while GWNN in the original vertex domain.

### 3.3  Visualization

In Figure 3, we illustrate the difference between GDN and each component inside by visualizing the reconstructed road occupancy rates in a traffic network. The traffic network targets District 7 of California collected from Caltrans Performance Measurement System (PeMS)[1]. We select 4438 sensor stations as the node set $V$ and collect their road average occupancy rates at 5pm on June 24, 2020. Following [23], we construct an adjacency matrix A by denoting $A_{ij} = 1$ if sensor station $v_j$ and $v_i$ are adjacent on a freeway along the same direction. We reconstruct road occupancy rate of each sensor station with three decoder variants: (b) inverse operator, (c) our proposed GDN, (d) GCN decoder [47]. Here the three variants share the same encoders.

As can be seen, while both GDN and GCN decoders can resemble the input signal in low frequency, GDN decoder retains more high frequency information. For inverse operator decoder, although keeping much high frequency information (mixed with noise), it drops lots of low frequencies, making the decoded signals less similar to the input in Euclidean space.

## 4  Experiments

We validate the proposed GDN on graph feature imputation [35, 44] and graph generation [19, 14]. The former task reconstructs graph features and the latter generates graph structures.

### 4.1  Graph feature imputation

In our model, we consider feature imputation as feature recovery on graphs. The model is trained on an incomplete version of feature matrix (training data) and further used to infer the potential missing features (test data).

**Datasets**  We use six benchmark datasets including several domains: citation networks (Cora, Citeseer) [32], product co-purchase networks (Amaphoto, Amacomp ) [26], social rating networks (Douban, Ciao[2]). For citation networks and product co-purchase networks, we use the preprocessed versions provided by [35] with uniform randomly missing rate of $10\%$. For Douban, we use the preprocessed dataset provided by [27]. For Ciao, we use a sub-matrix of 7,317 users and 1,000 items. Dataset statistics are summarized in Table 3 in the Appendix.

---

[1]http://pems.dot.ca.gov/
[2]https://www.ciao.co.uk/

Table 1: RMSE test comparison of different methods on graph feature imputation

| Datasets | Ciao | Douban | Cora | Citeseer | Amaphoto | Amacomp |
|---|---|---|---|---|---|---|
| MEAN | 1.385 | 0.850 | 0.503 | 0.503 | 0.414 | 0.417 |
| KNN | 1.461 | 0.817 | 0.459 | 0.443 | 0.421 | 0.422 |
| SVD [37] | 1.380 | 0.833 | 0.493 | 0.490 | 0.413 | 0.417 |
| MICE [4] | 1.600 | - | 0.480 | - | 0.481 | 0.487 |
| GAIN [43] | 1.099 | 0.809 | 0.433 | 0.421 | 0.420 | 0.416 |
| GRAPE [44] | 1.042 | 0.742 | 0.430 | 0.419 | 0.399 | 0.402 |
| Inverse decoder | 1.115 | 0.812 | 0.447 | 0.431 | 0.409 | 0.407 |
| GCN decoder[47] | 1.132 | 0.826 | 0.439 | 0.434 | 0.410 | 0.409 |
| GALA [30] | 1.147 | 0.833 | 0.457 | 0.435 | 0.415 | 0.411 |
| OURS | **1.011** | **0.734** | **0.415** | **0.399** | **0.391** | **0.393** |

**Setup**  Our graph autoencoder framework consists of an encoder and a decoder. Our encoder consists of two layers of Graph Convolutional Networks (GCN) [20]. Our decoder consists of one layer of proposed GDN, to produce fine-grained graphs. We use the Mean Squared Error (MSE) loss to reconstruct the features. We run 5 trials with different seeds and report the mean of the results. For detailed architecture, please refer to Appendix C. For detailed hyper-parameter settings, please refer to Appendix D. We report Root Mean Squared Error (RMSE).

**Baselines**  We compare with three commonly used imputation methods and three deep learning based imputation models:

- **MEAN**, which replaces the missing feature entries with the mean values of all observed feature entries.

- **KNN**, which first identifies the k-nearest neighbor samples and imputes the missing feature entries with the mean entries of these k samples. We set $k = 5$ in this work.

- **SVD** [37], which replaces the missing feature entries based on matrix completion with iterative low-rank SVD decomposition.

- **MICE** [4], which is a multiple regression method that infers missing entries by Markov chain Monte Carlo (MCMC) techniques.

- **GAIN** [43], which is a deep learning imputation model with generative adversarial training.

- **GRAPE** [44], which is the state-of-the-art deep learning imputation model based on deep graph representations.

For ablation studies of the graph autoencoder framework specified in Appendix C, we also include the results of decoders using GCN decoder [47], GALA [30], and inverse decoder in Section 3.2.1, to be compared with our proposed GDN. Besides, we compare with popular matrix completion methods in Appendix B for discrete feature recovery.

**Results**  The test RMSE on the six benchmarks are shown in Table 1. Our method performs the best on all datasets, e.g., it beats the second best model GRAPE by 0.02 on Citeseer and 0.031 on Ciao, which shows the superiority of our methods on feature imputation. The feature imputation RMSE with respect to different missing rate is shown in Figure 4. As can be seen, our method performs the best with respect to different missing rate, e.g., it achieves 0.016 improvement over the second best model GRAPE on Citeseer when the missing rate is 70%, and 0.007 improvement over the second best model GRAPE on Amacomp when the missing rate is 30%.

**Ablation study**  We investigate the role of wavelet de-noising in our GDN. We let the four decoder variants share the same depth of layers and parameter size. As observed in Table 1, when decoding by inverse operator, the performance is on a par with that of GCN decoders [47] and outperformed by our GDN decoders on all six datasets, indicating the effectiveness of this hybrid design. The performance of inverse decoder specified in Section 3.2.1 is always better than that of GALA [30], which shows the superiority of our third-order approximation.

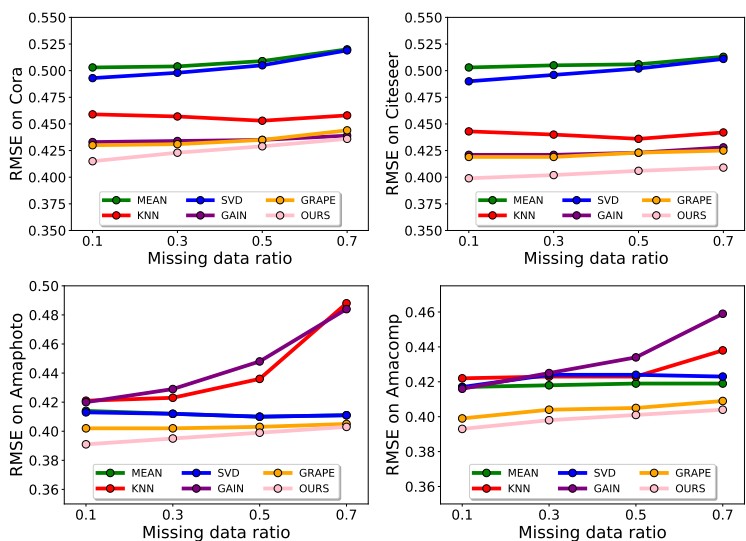

Figure 4: Comparison of different methods on graph feature imputation tasks. The x-axis denotes the missing ratio while the y-axis denotes RMSE on test set.

Table 2: The effect of GDN with various graph generation methods

| Datasets | MUTAG | | | PTC-MR | | | ZINC | | |
|---|---|---|---|---|---|---|---|---|---|
| - | $\log p(A\|Z)$ | AUC | AP | $\log p(A\|Z)$ | AUC | AP | $\log p(A\|Z)$ | AUC | AP |
| VGAE [19] | -1.101 | 0.716 | 0.427 | -1.366 | 0.566 | 0.433 | -1.035 | 0.556 | 0.288 |
| VGAE + GDN | **-1.026** | **0.823** | **0.611** | **-1.351** | **0.760** | **0.602** | **-1.006** | **0.858** | **0.611** |
| Graphite [14] | -1.100 | 0.732 | 0.447 | -1.362 | 0.564 | 0.437 | -1.039 | 0.553 | 0.288 |
| Graphite + GDN | **-1.024** | **0.818** | **0.608** | **-1.347** | **0.773** | **0.613** | **-0.998** | **0.838** | **0.567** |

## 4.2 Graph structure generation

**Data and baselines**    We use three molecular graphs: MUTAG [21] containing mutagenic compounds, PTC-MR [21] containing compounds tested for carcinogenicity and ZINC [17] containing druglike organic molecules, to evaluate the performance of GDN on graph generation. Dataset statistics are summarized in Table 4 in the Appendix. We test GDN on two popular variational autoencoder framework including VGAE [19] and Graphite [14]. Our purpose is to validate if GDN helps with the generation performance.

**Setup**    For VGAE and Graphite, we reconstruct the graph structures using their default methods and the features using GDN. The two reconstructions share the same encoders and sample from the same latent distributions. For detailed architecture, please refer to Appendix D. We train for 200 iterations with a learning rate of 0.01. For MUTAG and PTC-MR, we use $50\%$ samples as train set and the remaining $50\%$ as test set. For ZINC, we use the default train-test split.

**Results**    Evaluating the sample quality of generative models is challenging [36]. In this work, we validate the generative performance with the log-likelihood ($\log p(A|Z)$), area under the receiver operating characteristics curve (AUC) and average precision (AP). As shown in Table 2, GDN can improve the generative performance of both VGAE and Graphite in general, e.g., it improve the AP score of Graphite by $16.1\%$ on MUTAG , the AUC score of Graphite by $20.9\%$ on PTC-MR and $\log p(A|Z)$ of VGAE by $0.029$ on ZINC, which shows the superiority of GDN.

## 5    Related work

**Deconvolutional networks**    The area of signal deconvolution [22] has a long history in the signal processing community and is about the process of estimating the true signals given the degraded or smoothed signal characteristics [2]. Later deep learning studies [46, 29] consider deconvolutional

networks as the opposite operation for Convolutional Neural Networks (CNNs) and have mainly focused on Euclidean structures, e.g., image. For deconvolutional networks on non-Euclidean structures like graphs, the study is still sparse. [12] proposes the network deconvolution as inferring the true network given partially observed structure. [42] formulates the deconvolution as a pre-processing step on the observed signals, in order to improve classification accuracy. [47] considers recovering graph signals from the latent representation. However, it just adopts the filter design used in GCN and sheds little insight into the internal operation of GDN.

**Graph autoencoders**   Since the introduction of Graph Neural Networks (GNNs) [20, 6] and autoencoders (AEs), many studies [19, 14] have used GNNs and AEs to encode to and decode from latent representations. Although some encouraging progress has been achieved, there is still no work about graph deconvolution that can up-sample latent feature maps to restore their original resolutions. In this regard, current graph autoencoders bypass the difficulty via (1) non-parameterized decoders [19, 7, 24], (2) GCN decoders [14], (3) multi-layer perceptron (MLP) decoders [34], and (4) Laplacian sharpening [30].

**Graph feature imputation**   Feature imputation techniques can be generally classified into two groups: statistical approaches and deep learning models. The former group includes KNN, SVD [37], MICE [4], in which they makes assumptions about the data distribution through a parametric density function. The latter group includes GAIN [43], GRAPE [44]. Recently GRAPE [44] proposes graph feature imputation that assumes a network among items and the network is helpful in boosting the feature imputation performance. A related topic is matrix completion [27, 3, 11] in which it usually makes the assumption of finite and discrete feature values. Differently, both GRAPE and our method can recover both continuous and discrete feature values.

**Graph structure generation**   Graph structure generation techniques can be classified into three groups: one-shot [19, 14, 34], substructure [18] and node-by-node [25]. As this work is more related to the group of one-shot, we refer the readers to [10] for the details of other groups. The group of one-shot generates the entire graph all at once. Representative works include VGAE [19], Graphite [14] and GraphVAE [34]. Both VGAE and Graphite reconstruct graphs based on the similarities of graph structures. Differently, GraphVAE considers both the similarities of structures and node features and reconstructs node features using MLP. In this work, we show that the performance of one-shot graph generators can be improved by reconstructing node features using GDN.

## 6   Conclusion

In this paper, we present Graph Deconvolutional Network (GDN), the opposite of GCN that recovers graph signals from smoothed representations. The introduced GDN uses spectral graph convolutions with an *inverse* filter to obtain inversed signals and then de-noises the inversed signals in wavelet domain. The effectiveness of the proposed method is validated on graph feature imputation and graph structure generation tasks.

## 7   Limitation

Broadly speaking, graph signal restoration is an ill-posed problem. In this regard, the prior knowledge is important for high-quality recovery. This paper only pays attention to vanilla GCN [20] (Kipf & Welling, 2017) and it is not clear whether the spectral-wavelet GDN ideas can be applied to other GNNs, e.g., GAT [38], GIN [41]. Another limitation of this work is that the heat kernel in wavelet parts is not a band-pass filter, as it should be.

## 8   Broader Impact

This work initiates the restoration of graph signals smoothed by vanilla GCN [20] (Kipf & Welling, 2017). As a sequence, researchers in drug design or molecule generation may benefit from this research, since the visualization and understanding of GCN based representations is worthwhile to be further explored.

## Acknowledgments and Disclosure of Funding

The work was supported by NSFC Grant No. U1936205 and General Research Fund from the Research Grant Council of the Hong Kong Special Administrative Region, China [Project No.: CUHK 14205618].

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
