# OpenReview forum: "Deconvolutional Networks on Graph Data"
_NeurIPS.cc/2021/Conference — NeurIPS 2021 Poster_

### Official Review · Reviewer_J7HE · 2021-07-06

**Rating:** 6
**Confidence:** 3

**Summary:**

This paper presents Graph Deconvolutional Networks (GDN), a kind of graph neural network designed to invert the low-pass filtering operation implemented by Graph Convolutional Networks (GCN, Kipf & Welling, 2016).

GDN uses a high-pass filter to invert the effect of GCN and a learnable thresholding operation in the wavelet domain to remove any high-frequency noise.
The method relies on the Maclaurin series to approximate the high-pass filter (which in principle should be the exact inversion of GCN, although this is expensive) and the wavelet basis (which relies on the eigendecomposition of the graph Laplacian).

The authors conclude the paper with two main kinds of experiments:

1. Graph feature imputation, in which only a fraction of the nodes in a graph is observed and the remaining must be reconstructed.
2. Graph generation using graph variational autoencoders (Kipf & Welling, 2017) and the Graphite technique (Grover et al., 2019).

**Limitations And Societal Impact:**

The limitations are adequately addressed in a dedicated section and the paper is unlikely to have a negative societal impact.

**Main Review:**

**Originality**: the contributions of the paper are novel and clear, and related works are appropriately cited.

**Quality**: the paper is theoretically sound, with proofs included in the supplementary material. The derivation of the proposed methodology is clear and the authors clearly explain every choice that they have made (for example, the step at which to truncate the Maclaurin series is found experimentally).
The authors openly discuss the limitations of the work in a dedicated section.

**Clarity**: the paper is very well written and organized, guiding the reader through the math. Some prior knowledge of graph convolution is required to follow the paper, but this can be expected for this kind of work.
I am confident that I could reproduce the results of the paper given the description of the methodology.

**Significance**: the significance of the paper is limited.

- The main issue that I see is that there is no clear reason why a graph representation composed of only low and high frequencies should be more desirable than a graph representation of only low frequencies, especially for the tasks considered here.
The experiments mildly validate the authors' assumption but the performance improvements are not always significant w.r.t. prior art. The results can be just as well ascribed to the bigger number of parameters that the GNN has when using the GDN decoder, and it is not clear that the spectral analysis conducted to derive GDN is at all related to the performance improvements.
    - Related to the comment above, why do the GCN decoder and inverse decoder have the same performance in the first experiment, given that they have opposite behaviors? This likely indicates that the experiment is not strongly related to the analysis conducted in the paper.

- A second problem is that this method simply changes GCN from a low-pass to a band-stop filter, but there are classes of GCNs that can already implement such filtering operations, e.g., GCNs based on rational filters.

Overall, while the paper and the design of GDN are interesting, I don't think that they represent a major contribution to the field of GNNs.

**Comments**:

- The claim that the proposed method has a complexity of O(|E|) can be misleading because the method does not use the recursive implementation of the Chebynet paper, but rather pre-computes the powers of the Laplacian. This means that the cost of GDN will quickly converge to O(N^2) and will be O(N^2) by the time that n (index of the Maclaurin series) is as big as the graph diameter. In real-world networks and small graphs (like the ones tested in the paper), this number is likely to be small and therefore the cost of computing GDN is non-negligible.

- GDN is supposed to be an approximation of the exact inverse operator. However, from Fig. 3 we see that the two have qualitatively different behaviors in the low frequencies.
Why does GDN retrieve the low-frequency components whereas the exact inverse operator does not? Can the authors elaborate more on this aspect?

I am more than happy to revise my score if the authors address my concerns.

**Time Spent Reviewing:**

5

---

> ### Author Response · Authors · 2021-08-07
> **Response to Reviewer J7HE**
>
> We are most thankful for your thoughtful assessment, and glad to communicate with you on all your concerns:
>
> 1.	About the point 1 in Significance. We would like to highlight that our work is more related to graph reconstruction, rather than graph representation. We agree with you that it is doubtful a graph representation composed of low and high frequencies should be better than a representation of only low frequencies, especially for tasks like graph classification or graph clustering. However, when it comes to graph reconstruction, the same story does not apply. One can imagine graph signals as a piece of music in which low frequency signals correspond to low-pitched parts and high frequency signals to high-pitched parts. To classify different kinds of music, it might be good to only make use of low frequencies, in which the music style can be captured. Back to the cases in graph classification, there are deep reasons why low frequencies matter. As the algorithm of spectral clustering only requires low eigenvectors of graph Laplacian (see [1]), one can imagine that capturing low frequencies can capture the macro graph information (clusters), thus facilitating the classification tasks. However, if we want to reconstruct the original music, the high-pitched parts are important and cannot be disregarded as they involve lots of emotions and colors of the music. Back to the cases in graph reconstruction, the high-frequency signals might contain important and useful information. For example, in a social network the signals could be users’ gentle and a male and a female could become friends; if we encode the graph with GCN model, the derived user representations would be similar by retaining low-frequencies and suppressing high-frequencies; our GDN can thus be used to recover the original gentle information as much as possible (not perfectly at this moment as we have many restrictions such as efficiencies). In terms of the performance improvements, we cannot agree that the improvements are not always significant. As can be seen from Table 1, ours performs the best in feature reconstruction tasks across all six datasets. In terms of the question “why do the GCN decoder and inverse decoder have the same performance in the first experiment”, this shows that if we only inverse the kernel of GCN to reconstruct signals, we possibly result in a decoder that performs on par with GCN decoders, i.e., inverse decoder could reconstruct the high-frequency parts with the expense of inevitably amplifying the noise. This phenomenon shows that a de-noising component is in need for the inverse decoder. It is also one of our intuitions to design the hybrid spectral-wavelet reconstruction decoder.
> 2.	About the point 2 in Significance. We would like to clarify that our work is not to introduce a new GNN, in which the input graph structure and features are jointly transformed into graph representations. We agree that there are many GNNs, proposing a new one following previous paradigm does not represent a major contribution. Differently, in this work, we want to initial a new topic, i.e., $\text{given the graph representations, how to reconstruct the original input?}$ It can be seen as the inverse operation of GNNs. Since there are many GNNs, in our work, we target Vallina GCN and want to propose the inverse operation for Vallina GCN. As the kernel of Vallina GCN is $(1- \lambda)$, a low-pass, naively inversing the kernel would result in $\frac{1}{1-\lambda}$, a high-pass, which could amplify graph noise. We thus introduce a de-noising component using wavelet (which in perfect situation should be a band-pass filter). In other words, our intuition is not to change GCN from low-pass to a band-stop filter, as discussed in your comments. Instead, given GCN is a low-pass filter, how can we recover as much information as possible with the limitation of efficiency and noise involved. We agree that this work can be improved, e.g., the current wavelet network reduces the noise at the expense of dropping lots of high-frequencies, as can be seen in Fig 3.c. As a starting work, we think this work should be a reference for following researches in graph reconstruction areas for these foundings: 1. naively inversing the kernel of GCN to recover the graph signal may amplify noise and is not optimal; 2. the trade-off between reducing noise and retaining useful information should be considered and a better de-noising component is in need.
> 3.	About the point 1 in Comments. Very good discussions here. In fact, the complexity should be $O(\sum d_i^n)$ where $d_i$ is the degree for node $i$ and $n$ is the index of the Maclaurin series. It is dominated by the index $n$, i.e., if $n$ is large as discussed in your comments, the cost grows very quickly, approaching $O(N^2)$. In this context, we limit $n$ to be $n \leq 3$, as discussed in line 112 of the paper. Since $d_i^3$ follows a Power Law distribution, the complexity is dominated by these nodes with the largest degrees. In the implementation, we set a threshold to make sure $d_i^3$ is smaller than a given constant. Thus, the complexity is still on par with $O(E)$. In fact, many works including GWNN use this thresholding methods to ensure the powers of the Laplacian do not go from sparse to dense in their implementations, though they don’t discuss this point explicitly. We agree this complexity analysis could be improved and will give more detailed discussion in the revision.
> 4.	About the point 2 in Comments. If the de-noising component, i.e., the wavelet network, works perfectly, GDN is supposed to act similarly with the exact inverse operator, with some minor differences of noise reduction. However, when using wavelet to do noise reduction, we have some restrictions. Firstly, following previous works, we use heat kernels in wavelet designs. But according to [2], the wavelet kernel should be a band-pass filter. Obviously heat kernel is not a band-pass filter. Here one simple idea is that we can replace the heat kernel by some band-pass filters such as Haar wavelet. However, as Haar wavelet is not smooth and not differentiable, we immediately meet with another problem: the polynomial approximation becomes useless and we have to rely on explicitly eigen-decompositions, which is really expensive to compute. In short, the reason why we choose heat kernel is a tradeoff between efficiency and effectiveness. In fact, from Eq. 13 one can understand the heat kernel-based wavelet as a high-pass GNN followed by a low-pass GNN. As the last layer is a low-pass GNN, it is imaginable that this de-noising component will somehow drop high-frequencies, which makes GDN act not that similar with the exact inverse operator. Secondly, in each of the neural layers, we apply non-linear activation functions such as ReLU, which makes the whole system highly non-linear. In summary, the main purpose of GDN is to reconstruct as much original features as possible, rather than an approximation of the exact inverse operator. From Fig.3, we can see that GDN reconstructs better than both GCN and inverse operator. From this point of view, our reconstruction purpose is achieved, though we definitely have spaces to improve our work in the future.
>
> [1] Ulrike Von Luxburg. “A tutorial on spectral clustering”.\
> [2] Hammond et al. “Wavelets on graphs via spectral graph theory”.

---

> > ### Comment · Reviewer_J7HE · 2021-08-27
> > **Raised score**
> >
> > I thank the authors for their patience, I wanted to make sure that I understood the authors' response and I took the time to read the paper again before answering.
> >
> > I think that the authors have successfully addressed my concerns and I see that they have also performed new experiments that overall make the paper stronger.
> >
> > I have therefore raised my score from a 5 to a 6.

---

> > > ### Author Response · Authors · 2021-09-02
> > > **RE:Raised score**
> > >
> > > Dear Reviewer J7HE,
> > >
> > > We thank you for your cautious and insigntful assessment. We think there is a mutual agreement between us that this work addresses an important but scarcely explored problem -- the definition of an efficient and effective Deconvolutional Network for Graph data. We are encouraged by your positive comments and will put more efforts on this direction in future works.
> > >
> > > All the best, Authors

---

> ### Author Response · Authors · 2021-08-27
> **The end of the discussion phase approaching**
>
> Dear Reviewer J7HE,
>
> We consider your assessment highly important and helpful. As the end of the discussion is approaching, we really appreciate if you could go over our response and let us know if there is anything else we should address. Thank you so much again for your time and effort to make our work better.
>
> All the best,
> Authors

---

### Official Review · Reviewer_QTzU · 2021-07-15

**Rating:** 6
**Confidence:** 4

**Summary:**

This paper proposed GDN, the opposite of GCN that recovers graph signals from smoothed representations. The introduced GDN uses spectral graph convolutions with a high pass filter to obtain inversed signals and then de-noises the inversed signals in wavelet domain. Effectiveness of GDN is validated on graph feature imputation and graph structure generation tasks.

**Limitations And Societal Impact:**

Yes.

**Main Review:**

Overall, this work is a clearly-structured paper. I think it is about average. For the following reasons:

1) Techniques:
Overall, techniques proposed in this paper seems reasonable.
One problem is that there is analysis on white and Gaussian noise, analysis on other types of noise is expected to be given.

2) Experiments

2.1)	This paper lacks general tasks, such as node classification and link prediction tasks, when compared with GNNs, such as Graphsage and GAT. In highly related works, such as [1] also provides node classification results to show the effectiveness of proposed model.

2.2)	This paper claims that proposed model is applicable to continuous and discrete feature values. However, results in Table 5 and Table 3 are all datasets with discrete values. Results on continuous feature datasets seem lacked.

3) Writings:
The paper is clearly-structured and easy to follow, definitions of symbols are clear.

[1] Handling Missing Data with Graph Representation Learning. NeurIPS, 2020.


**Time Spent Reviewing:**

12hours

---

> ### Author Response · Authors · 2021-08-10
> **Response to Reviewer QTzU**
>
> We sincerely appreciate your positive comments that the technical part is concrete and clear, and delighted to address all your concerns:
>
> 1.	About other types of noise. Good suggestions. Our current noise analysis framework does rely on the rotation invariant property (Fact 2), which plays a crucial role in quantifying the error (i.e., the gap between the ground truth and the noisy one). A simple example which statisfies the rotation invariant property is Gaussian, which has been included in our paper. Besides that, slightly less-immediate examples would be uniform distribution over the sphere, see Chapter 4 in [11]. However, it is worth noting that other analysis frameworks can possibly handle other types of noise. We leave it as a future work.
>
> 2.	About lacking general tasks such as graph classification. Very constructive idea. We would like to clarify three points firstly:
>
> *  Our proposed GDN is to reconstruct graph representations smoothed by Vallina GCN. The derived graph representations are in an unsupervised fashion and $\textit{can}$ be used for node classification and graph classification tasks, if followed by classifiers.
>
>  * It is $\text{doubtful}$ reconstruction-oriented representations are more suitable than proximity-oriented representations (e.g., mutual information) in terms of classification tasks. The reason is simple, for the purpose of reconstruction, the representations should contain as much components of the original input as possible; while for the purpose of classification, the representations need to embed the discriminative components in order to have a margin between classes. A more detailed discussion can be found at the response to Reviewer J7HE (item 1).
>  * In GRAPE [1], the node classification task (named as label prediction) is not a typical node classification task, i.e., all the methods first impute the features and then use linear regression classifier to do node classification. In other words, the main purpose of label prediction in GRAPE is to evaluate their abilities to recover the features. This point is obvious as almost all the baselines of label prediction in GRAPE are imputation methods, and there are no GNN baselines such as Graphsage and GAT.
>
> Nevertheless, motivated by your suggestions, we test the capacity of GDN in graph classification with the following setting. We use a graph autoencoder framework to derive graph representations and report the mean 10-fold cross-validation accuracy using LIBSVM. Our encoder consists of two layers of Vallina GCN and a pooling layer [8]. Our decoder consists of an unpooling layer [9] and one GDN layer. We adopt the same procedure of previous works [6,7] and use five graph classification benchmarks including IMDB-Binary, IMDB-Multi, Reddit-Binary, PROTEINS and DD. We compare with WL graph kernels [2] and five unsupervised graph-level representation learning methods: VGAE [3], SUB2VEC [4], GRAPH2VEC [5], INFOGRAPH [6] and MVGRL [7].
>
> |               | IMDB-BIN | IMDB-MULTI | REDDIT-BIN | PROTEINS | DD   |
> |---------------|----------|------------|------------|----------|------|
> | WL [2]        | 72.3     | 47.0       | 68.8       | 72.9     | 76.4 |
> | VGAE [3]      | 64.9     | 38.9       | -          | 72.4     | 76.3 |
> | SUB2VEC [4]   | 55.3     | 36.7       | 71.5       | -        | -    |
> | GRAPH2VEC [5] | 71.1     | 50.4       | 75.8       | 73.3     | -    |
> | INFOGRAPH [6] | 73.0     | 49.7       | 82.5       | 75.0     | 76.9 |
> | MVGRL [7]     | 74.2     | 51.2       | 84.5       | 75.9     | 78.3 |
> | GDN           | $\text{76.0}$     | $\text{51.5}$       | $\text{86.0}$       | $\text{76.1}$     | $\text{79.1}$ |
>
> \* Note here we don't compare with baselines such as GIN/GAT where label information is utilized in the training process.
>
> 3.	About continuous graph features. In the visualization part Figure 3, GDN is used to recover road occupancy rate, which is continuous. To have a concrete comparison with previous works, we add one more dataset—OGB-arxiv [10], which has 169,343 nodes and whose node features are continuous. We randomly split the data and use 70% node features as train set and the remaining 30% as test set. The RMSEs of MEAN, KNN, SVD, MICE, GAIN, GRAPE and GDN are 9.72%,9.24%,8.36%,8.91%,8.75%,8.73% and $\text{8.21}$% respectively. From these results, one can see GDN performs the best for continuous feature imputation.
>
> [1] Handling Missing Data with Graph Representation Learning. NeurIPS, 2020.\
> [2] Weisfeiler-lehman Graph Kernels. JMLR, 2011.\
> [3] Variational Graph Auto-encoders. NeurIPS-W, 2016.\
> [4] Sub2vec: Feature Learning for Subgraphs. PAKDD, 2018.\
> [5] Subgraph2vec: Learning Distributed Representations of Rooted Sub-graphs from Large Graphs. 2017.\
> [6] Infograph: Unsupervised and Semi-supervised Graph-level Representation Learning via Mutual Information Maximization. ICLR, 2020.\
> [7] Contrastive Multiview Representation Learning on Graphs. ICML, 2020.\
> [8] Hierarchical Graph Representation Learning with Differentiable Pooling. NeurIPS, 2018.\
> [9] Spectral Clustering with Graph Neural Networks for Graph Pooling. ICML, 2020.\
> [10] Open Graph Benchmark: Datasets for Machine Learning on Graphs. NeurIPS, 2020.\
> [11] Normal Distribution Characterizations with Applications.

---

> > ### Comment · Reviewer_QTzU · 2021-08-25
> > **Response to authors**
> >
> > I would like to thank the authors for providing a detailed response to my comments. My concerns are dispelled.

---

> > > ### Author Response · Authors · 2021-08-26
> > > **RE:Response to authors**
> > >
> > > Dear Reviewer QTzU, we are most glad to see your concerns are dispelled. If you have any follow-up questions, we are pleased to answer them and further nurture this deep graph reconstruction topic.

---

### Official Review · Reviewer_awFr · 2021-07-16

**Rating:** 7
**Confidence:** 3

**Summary:**

In this paper the authors introduce a Graph Deconvolutional Network (GDN) as a combination of inverse GCN filters (spectral domain) and denoising layers (wavelet domain). They motivate the design of their GDN by observing that a simple inverse GCN filter likely results in noise amplification.
The proposed architecture also has computational and scalability advantages over the previous ones in the literature.

The authors test the capabilities of the GDN on two tasks:
1. feature imputation (reconstructing graph *features*)
2. graph generation (reconstructing graph *structure*), where GDN is used to enhance the performance of two graph-generation techniques

They compare their proposed architecture with various baseline and state-of-the-art approaches and achieve the best performance in all of the tasks.

**Limitations And Societal Impact:**

The authors have clearly addressed the limitations of their method and suggested directions for further research.

They have briefly introduced possible negative societal impacts, but it might be adequate given the generality of the method introduced.

**Main Review:**

The paper tackles a new, scarcely explored problem: the definition of an efficient and effective Deconvolutional Network for Graph data.

It seems to be not the first paper to propose a definition / implementation for a GDN (see [50] Zhang et al.). The de-noising strategy is also borrowed by one of the cited works.

However, they seem to combine the ideas to create more efficient and effective version of a GDN, moving forward the discussion on how to better face this problem.
They also:
- openly and clearly point out the differences between their work and the related ones
- provide an in-depth explanation of their design choices
- compare the performance of all the related methods on all the tasks

---

The submissions appears technically sound. All claims are supported by proofs provided in the Appendix or by experimental results.

However, I would have found the results to be more solid if they were averaged over multiple runs (especially for the first task)

---

The paper is clearly written, well organized, and provides clear and on-point explanations of the differences between the proposed method and the ones it is compared against.

Some typos / confusing parts:
- use of the word *arts* (e.g. line 24, 28) instead of *articles*
- caption of Figure 2: *s=1* $\rightarrow$ *s=3* (maybe)
- line 121, wrong figure  *Figure 2 (b)* $\rightarrow$ *Figure 3 (b)*
- line 287,291: citation *(Kipf & Welling, 2017)* should be formatted as number
- line 224: capitalize *The performance*
- Confusing use of the term *inverse operator*.
  -  at line 119 you title the paragraph "Inverse operator vs [50]", meaning that "inverse operator" is your proposed GDN (minus the de-noising part)
  - in Figure 3/Section 3.3, "inverse operator" refers to [50] instead (if I understand correctly)
  - but [50] is also called "GCN decoder" in Table 1

  Please use consistent naming, it is really difficult to follow.

---

The submission presents a well justified and well documented improvement on the GDN model, showing practical improvements over the current baselines on a selection of tasks and datasets. The paper and related additional materials can be of help to further the research on this topic.

---

**After Author Response**

The authors addressed all my concerns and I confirm my initial evaluation.

**Time Spent Reviewing:**

8

---

> ### Author Response · Authors · 2021-08-09
> **Response to Reviewer awFr**
>
> We are sincerely grateful that you have highlighted many core ideas and main contributions of this work, and very glad to address all your concerns:
>
> 1.	About multiple runs for the first task. Good suggestions. We fully agree with you that it is more convincing that the experiment is done multiple times with different initial random seeds, as the results of many neural network methods are sensitive to the initial seeds. In this regard, for all experiments in this paper, we indeed run 5 trials with different seeds and report the mean of the results, following the settings in [1]. We shall update our settings in the revision. Motivated by your suggestions, we feel it is also our duty to report the Standard Deviation (SD) of the results here. as in the following table. From these results, one can see that deterministic models such as MEAN, KNN and SVD are not sensitive to initial seeds; another observation is that the SDs of GRAPE and GDN are approximately the same across all datasets.
>
> |       | Ciao        | Douban      | Cora        | Citeseer    | Amaphoto    | Amacomp     |
> |-------|-------------|-------------|-------------|-------------|-------------|-------------|
> | MEAN  | 1.38 ± 0    | 0.85 ± 0    | 0.50 ± 0    | 0.50 ± 0    | 0.41 ± 0    | 0.41 ± 0    |
> | KNN   | 1.46 ± 0    | 0.81 ± 0    | 0.45 ± 0    | 0.44 ± 0    | 0.42 ± 0    | 0.42 ± 0    |
> | SVD   | 1.38 ± 0    | 0.83 ± 0    | 0.49 ± 0    | 0.49 ± 0    | 0.41 ± 0    | 0.41 ± 0    |
> | MICE  | 1.60 ± 0.04 |    -        | 0.48 ± 0.01 |    -           | 0.48 ± 0.03 | 0.48 ± 0.04 |
> | GAIN  | 1.09 ± 0.03 | 0.80 ± 0.02 | 0.43 ± 0.01 | 0.42 ± 0.03 | 0.42 ± 0.04 | 0.41 ± 0.03 |
> | GRAPE | 1.04 ± 0.02 | 0.74 ± 0.01 | 0.43 ± 0.01 | 0.41 ± 0.03 | 0.39 ± 0.02 | 0.40 ± 0.02 |
> | GDN   | 1.01 ± 0.01 | 0.73 ± 0.01 | 0.41 ± 0.01 | 0.39 ± 0.03 | 0.39 ± 0.02 | 0.39 ± 0.02 |
>
> \* Note: while there are other multiple run methods such as the number of clusters in KNN and the rank in SVD, in this table we pay only attentions to the effects of random seeds.
>
> 2.	About the terms used in inverse operator. Very good point. We agree that we could make it better by using consistent naming. In terms of “inverse operator is your proposed GDN (minus the de-noising part)”, yes, this is correct, inverse operator is firstly proposed in this work, though it has the noise issue. In terms of “in Figure 3, inverse operator refers to [50]”, a simple answer to this is no, the reference [50] uses GCN as decoders and is used interchangeably with GCN decoder in current version; in fact, the Figure 3 is to illustrate the differences among GCN decoder [50], the inverse operator and GDN. In terms of “but [50] is also called GCN decoder”, yes, this is correct. In summary, we consider we can improve this naming issues by (1) using GCN decoder [50] wherever GCN decoder or [50] appears, and (2) explicitly explaining that GCN decoder and [50] are the same in line 119.
> 3.	About typos. In terms of the word arts, we shall change it into articles in the revision, in case there is any ambiguity. In terms of caption of Figure 2 and s=1, it is actually not a typo, and s here denotes a scaling parameter rather than Maclaurin series. For Figure 2 (b), it is a typo and should be Figure 3(b). For citation GCN (Kipf & Welling, 2017), we shall change it to GCN [21] (Kipf & Welling, 2017), for the purpose of emphasizing the specific variant of GNNs we are addressing. In terms of “The performance”, yes, it is a typo. We shall update these points in the revision.
>
> [1] Handling Missing Data with Graph Representation Learning. NeurIPS, 2020.

---

> > ### Comment · Reviewer_awFr · 2021-08-28
> > **Response to Authors**
> >
> > I thank the authors for the in-depth response.
> >
> > 1. I am glad to know that the results are actually averaged over different random seeds and not the result of a single run. I think it would be great to add it explicitly in the paper. The low standard deviations are also a really good indicator of the soundness of the results.
> >
> > 2. Thank you for clearing up the confusion: the proposed changes should be able to improve the readability.
> >
> > 3. I see, I confused $n=3$ (in the text) with $s=1$ in the caption of Figure 2. Thanks for taking the time to respond to all the points.

---

### Official Review · Reviewer_Vnte · 2021-07-20

**Rating:** 6
**Confidence:** 4

**Summary:**

This paper derives an inverse operator, which is said to be better than other methods in ref [50] and GALA. The work also points out the inverse operation results in a high pass filter and may amplify the noise. Motivated by this observation, a de-noising layer is designed and introduced into the proposed network. The graph deconvolution uses the wavelet transforms on the graph, which is implemented based on heat kernel and the series in terms of symmetric graph Laplacian. Theoretical analysis was provided for the derivation of the deconvolutional layer.

**Limitations And Societal Impact:**

It is unclear to me where the low-pass and high-pass information are used in the proposed GDN model. In fact, from (10)--(13), it is difficult to tell where the multiscale property of the transform $\Psi_s$ is employed.

It would be much more persuasive if the authors could provide a toy example that showcases the method's computational speed or test some real large datasets, such as open graph benchmarks (OGB).

**Main Review:**


* Figure 3 is not clear to observe useful information for illustrating the performance comparison for different approaches.

* How to efficiently compute the series in (11) and (12), which are used in calculating the GDN convolution in (13)? If there is a truncation, what is a suitable degree?

* OGB should be considered to test by the proposed GDN and compared with existing methods.

* In the experiment part, there are three different tasks: imputation and graph generation. The author evaluated the graph generation results with AUC score, AP score and the negative log-likelihood of the latent variable posterior distribution. There is no standard deviation of graph generation's performance like graph classification experiment. It is hard to reproduce the model if the standard deviation is too high.

* Would it be appropriate to put section 3.3 in the experiment part as an example of the model utilizing the high pass signal?

* line 48, typo for 'de-nosing'

* The notation for the graph is usually defined as the triplet of $V, E,w$, which represent vertex set, edge set and edge weights, respectively.

* line 73, 'We show' to 'we show'

**Time Spent Reviewing:**

5

---

> ### Author Response · Authors · 2021-08-10
> **Response to Reviewer Vnte**
>
> Thanks for your comments. We think the assessment is mixed, with some  misunderstanding and constructive points. We shall firstly address these misunderstanding, as follows:
> >In the experiment part, there are three different tasks: graph classification, social recommendation, graph generation.
>
> This is a critical misunderstanding. The fact is that we test our model in two tasks: feature imputation and graph generation, as raised by all the three other reviewers. We think the experiment is one of the cores in NeurIPS submissions. Mistaking this part would make the assessment inaccurate.
> >Appendix A, as mentioned in the main text, is not seen.
>
>  This is not correct. The fact is that Appendix A is right there in Supplementary Material. In particular, Reviewer awFr explicitly mentions that "All claims are supported by proofs provided in the Appendix". As another point, Reviewer J7HE says "the paper is theoretically sound, with proofs included in the supplementary material". All these evidence shows that the Supplementary Material can be found easily and is important to our work.
>
> -------------------------------------------------------------------------
> Meanwhile, we think other comments are relevant to our work, and glad to address them as follows:
>
> > Figure 3 is not clear to observe useful information for illustrating the performance comparison for different approaches.
>
> Figure 3 is used to give an intuitive understanding of the differences of various decoders including Vallina GCN (used in many works such as Graphite), inverse operator (naively proposed in this work and having GALA as its special case), and GDN. One simple observation is that GDN can better reconstruct high-frequencies than Vallina GCN if we put the two side by side. For the comparison between  inverse operator and GDN, if we zoom in, we shall find that GDN can better recover the road occupancy rates in Eculidean space; a possible confusing thing here is that it is hard to compare inverse operator and GDN in spectral domain, as inverse operator seems having better performance in high-frequencies but worse performance in low-frequencies.  The reasons are:
> * The heat kernel used in wavelet de-noising part is not perfect. From Eq. 13 one can understand the heat kernel-based wavelet as a high-pass GNN followed by a low-pass GNN. As the last layer is a low-pass GNN, it is imaginable that this de-noising component will somehow drop high-frequencies, which makes GDN act not that similar with the inverse operator.
> * In each of the neural layers, we apply non-linear activation functions such as ReLU, which makes the whole decoder highly non-linear.
>
> For a detailed discussion of why inverse operator and GDN act not that similar, please refer to the response to Reviewer J7HE (item 4). In short, Fig 3 is informative, as one can intuitively understand  that (1) Vallina GCN decoder would make the reconstructed features over-smoothed and hence not appropriate, (2) inverse operator decoder reconstructs well in high-frequencies and badly in low-frequencies, and (3) while GDN performs better than Vallina GCN decoder and inverse operator, it can be improved in the future.
>
> >How to efficiently compute the series in (11) and (12), which are used in calculating the GDN convolution in (13)? If there is a truncation, what is a suitable degree?
>
> We limit $n$ to be $n \leq 3$ where $n$  is the index of the Maclaurin series, as discussed in line 112 of the paper. Specifically, we have discussed the effect of orders of Maclaurin Series in GDN on the performance of graph structure generation in Table 7 of the appendix.
>
> > OGB should be considered to test by the proposed GDN and compared with existing methods.
>
> We add one more dataset—OGB-arxiv, which has 169,343 nodes and whose node features are continuous. We randomly split the data and use 70% node features as train set and the remaining 30% as test set. The RMSEs of MEAN, KNN, SVD, MICE, GAIN, GRAPE and GDN are 9.72%,9.24%,8.36%,8.91%,8.75%,8.73% and 8.21% respectively. From these results, one can see GDN performs the best on OGB dataset.
>
> >Would it be appropriate to put section 3.3 in the experiment part as an example of the model utilizing the high pass signal?
>
> Very constructive idea. On this road occupancy rate recovery experiment, the RMSEs of MEAN, KNN, SVD, MICE, GAIN, GRAPE and GDN are 0.24,0.21,0.20,0.17,0.15,0.10 and 0.07 respectively. We shall add the results in the revision.
>
> > line 48, typo for 'de-nosing'.
>
> Thanks. It is indeed a typo.
>
> >The notation for the graph is usually defined as the triplet of $V, E, w$, which represent vertex set, edge set and edge weights, respectively.
>
> We don't think we need to change notations, given the notations are clear and consistant. As another perspective, Reviewer QTzU explicitly mentions  "the definitions of symbols are clear".
>
> >line 73, 'We show' to 'we show'.
>
> Thanks. It is a typo and we shall correct it in the revision.

---

> > ### Comment · Reviewer_Vnte · 2021-09-02
> > **Almost solved my questions**
> >
> > The authors solved most of the questions. For Figure 3, I mean the resolution of the picture can be improved. It is a bit blurring and could be improved for example by using another example.

---

> > > ### Author Response · Authors · 2021-09-02
> > > **RE:Almost solved my questions**
> > >
> > > Dear Reviewer,
> > >
> > > We are very glad to know that most of your questions have been well resolved. In terms of the resolution of Figure 3, we highly appreciate this point and will replace it with a higher resolution one in the revision. If we understand it correctly, you are satisfied with our response and acknowledge our contributions. We will be very happy to know  if you could change the score accordingly. Thank you again!
> > >
> > >
> > > All the best,
> > > Authors

---

> ### Author Response · Authors · 2021-09-02
> **The discussion will be closed today**
>
> Dear Reviewer Vnte,
>
> We highly appreciate your comments and would like to take the last opportunity to address your follow-up concerns, as the discussion will be closed today. We have made all our efforts to respond your reviews and not received your response yet. We sincerely thank you for your time and efforts in reviewing our paper.
>
> All the best,
> Authors

---

### Author Response · Authors · 2021-08-25
**The end of the discussion phase approaching**

Dear Reviewers,

Could you please go over our responses since the end of the discussion phase is approaching? We have responded to your comments, openly discussed many aspects in areas of deep graph reconstruction, and provided additional experimental results that you have requested. We sincerely thank you for your time and efforts in reviewing our paper, and your insightful and constructive comments.

Thanks, Authors

---

### Decision · Program_Chairs · 2021-09-27

**Decision:**

Accept (Poster)

**Comment:**

The paper proposes a (niche) topic that is still not well explored, i.e. the definition of an efficient and effective deconvolutional network for graph data. This is done by mainly exploiting already existing results, however introducing a significant level of novelty. During the discussion period the authors were able to clarify all the main issues raised by the reviewers.  An additional contribution to the evaluation of the merits of the paper was coming from another AC, that although recognising the technical correctness of the approach, pointed out some issues that can be summarised as follows: i) motivation of the work is weak since a clear need for GDN and a target application is not clearly identified; ii) proposed experiments do not go to the core point, both in  terms of used datasets (in Cora and Citeseer lowpass information is generally sufficient for the classification task), and more importantly - in terms of the methods GDN is compared to; in short, the experimental assessment has been adapted to search for a justification of the definition of "the opposite of GCN"; iii) since at the end of the day the main component in the proposed method is the combination of lowpass and highpass filters, essentially to formulate bandpass filters that capture sufficient information for imputation and graph generation, it would be appropriate to compare the method to other graph filter constructions: in addition to the architectures based on Chebyshev polynomials (e.g., Chebynet) mentioned by the authors, numerous highly-related works have studied the use of various filter constructions to retain and recover highpass and bandpass information, including CayleyNets (Levie et al. 2017), MixHop (Abu-El-Haija et al., 2019), interferometric graph transforms (Oyallon, 2020), GNNs with ARMA filters (Bianchi, 2021), and some extensive body of work on a variety of graph scattering architectures (for example, Gama et al. 2018; Gao et al. 2019; Zou & Lerman 2020; Ioannidis et al. 2020); these works all aim to leverage high pass, bandpass, or wavelet filters to extract information beyond the low frequencies retained by the Kipf & Welling GCN, with numerous applications considered, which would provide a more established benchmark for the proposed method; in particular, for graph generation, [Zou & Lerman (2019)](https://doi.org/10.1109/IJCNN.2019.8851705) and [Castro et al. (2020)](https://ieeexplore.ieee.org/document/9378305) both showed the efficacy of scattering networks in such tasks.
In summary, the contribution seems to be technical correct, however it is not striking due to the direct exploitation of already published results, it needs a better motivation, consequent evaluation, and presentation. Notwithstanding these issues, that should be addressed by the authors, the proposed approach could eventually be used for works in explainability, which is a very important direction of research to aim for.